# Increasing Care Partners’ Capacity for Supporting Individuals Living with Dementia Through Bravo Zulu: Achieving Excellence in Relationship-Centered Dementia Care

**DOI:** 10.3390/ijerph22070970

**Published:** 2025-06-20

**Authors:** Jennifer Carson, Taniya J. Koswatta, Samantha Hoeper, Peter S. Reed

**Affiliations:** 1Dementia Engagement, Education, and Research (DEER) Program, School of Public Health, University of Nevada, Reno, NV 89557, USA; jennifercarson@unr.edu; 2Department of Health Behavior, Policy and Administration Sciences, School of Public Health, University of Nevada, Reno, NV 89557, USA; shoeper@unr.edu (S.H.); psreed@unr.edu (P.S.R.); 3Sanford Center for Aging, School of Medicine, University of Nevada, Reno, NV 89557, USA

**Keywords:** dementia care training, personhood beliefs, self-efficacy

## Abstract

The need for person- and relationship-centered care (PCC/RCC) in Alzheimer’s disease and other dementias is well established. Recognizing the limitations of PCC in fully honoring the intricate interdependencies between care partners and persons living with dementia, a new training program called Bravo Zulu was developed. This comprehensive, 12-hour dementia training program aims to enhance personhood beliefs and self-efficacy among care partners, improving the experience of care and support for both people living with dementia and their care partners. Responses from 182 participants who completed the training were analyzed using paired t-tests to assess changes in personhood beliefs and self-efficacy. The Bravo Zulu training produced significant increases in both personhood beliefs and self-efficacy. Notably, healthcare professionals without prior care partner training exhibited the greatest gains in personhood beliefs, while participants who were not direct care partners showed substantial improvements in self-efficacy. Overall, these findings support the concept of tailoring dementia education to ensure care partners and healthcare professionals are able to provide culturally competent care that is aligned with the diverse backgrounds of people living with dementia. Expanding access to high-quality interactive programs such as Bravo Zulu can contribute to strengthening the dementia care workforce and improving care experiences for all involved.

## 1. Introduction

In the United States, approximately 7 million individuals aged 65 or older live with Alzheimer’s disease or another form of dementia [1]. As dementia progresses, the need for person- and relationship-centered approaches to care and support becomes increasingly evident [2]. Despite these efforts, people living with dementia (PLWD) often experience a loss of meaning and connection in relationships, contributing to distress and disconnection [3,4]. At the same time, caregivers—here reframed as “care partners”—face heightened health risks, emotional and financial strain, and job dissatisfaction [5,6,7,8]. Research underscores that meaningful relationships between PLWD and their care partners are essential for quality care, well-being, and job satisfaction [9,10]. This suggests that fostering reciprocal relationships within dementia care is not only beneficial but crucial for the health and well-being of both PLWD and their care partners.

Building on the foundation of person-centered care (PCC), there is growing recognition of the need for relationship-centered care (RCC) in dementia care settings [11]. The Institute of Medicine [12] defines PCC as being “respectful of and responsive to an individual’s preferences, needs, and values and ensuring that the individual’s values guide all clinical decisions” (p. 6). Similarly, the Centers for Medicare & Medicaid Services define PCC as “integrated health care services delivered in a setting and manner that are responsive to individuals and their goals, values, and preferences, in a system that supports good provider–patient communication and empowers both individuals receiving care and providers to make effective care plans together” [13]. Within both definitions, we see an emphasis on a person’s preferences and values, which are often deeply shaped by their culture, making cultural competency and cultural humility essential to delivering PCC. Further, there is an emphasis on the dyadic dynamic informing care-related decision making, imbuing the goal of person centeredness with a direct focus on the care relationship [14]. While PCC rightfully prioritizes the individual’s values and preferences [2,15], it primarily considers care as something provided to a person rather than something that occurs within dynamic, reciprocal relationships [16]. However, dementia care does not happen in isolation—it unfolds within a web of relationships between PLWD, care partners, and the broader community. Recognizing this dynamic, there is growing acknowledgment that an RCC approach is necessary to fully support well-being in dementia care settings. RCC builds upon the principles of PCC but by expanding the focus to emphasize the interdependent nature of care relationships and their impact on health, well-being, and quality of life [17]. As Nolan et al. [11] describe, optimal dementia care occurs when all individuals in the care context experience relationships that promote the following ‘senses’:Security—feeling safe within relationships;Belonging—feeling part of something meaningful;Continuity—experiencing connection and stability;Purpose—having personally meaningful goals;Achievement—making progress toward desired goals;Significance—feeling that you matter [11] (p. 49).

While PCC has demonstrated positive impacts on depression and quality of life for PLWD, it does not fully account for the interdependent nature of care relationships [15,18,19,20]. Care partners play a crucial role in shaping the well-being of PLWD, and their own experiences of ill-being or well-being can directly affect the quality of care and support they provide [2,21]. A truly holistic approach must recognize the reciprocal nature of well-being—when care partners thrive, PLWD are more likely to experience compassionate, person- and relationship-centered support and also thrive. Furthermore, PCC approaches often rely heavily on the observations and decision-making of care partners, which are shaped not only by their knowledge of the person they support, but also by their own cultural backgrounds and perceptions [22]. Without intentional efforts to develop cultural competency and humility, care partners may unintentionally impose their own values and assumptions onto PLWD, rather than honoring each individual’s unique identity and values in care decisions [23]. RCC provides a framework to address these gaps, ensuring that care is not only person-centered but also relationally responsive and culturally attuned.

Too often, well-intended but paternalistic PCC models position PLWD as passive recipients or “objects of care” rather than as partners and legitimate contributors within the circle of care [24]. This observer-centric ‘caregiver’ approach risks pathologizing expressions of distress by labeling them as so-called “behaviors” rather than recognizing them as communicative expressions of unmet needs. Without empathetic curiosity, cultural humility, relational care, and active dialogue, care decisions—including so-called “behavioral interventions”—may overlook the values, preferences, unique needs, and lived experiences of PLWD [14].

In response to these limitations, there is a growing demand for dementia education and training programs that cultivate dementia-capable, person- [and relationship-centered] workforces in the United States [25]. Such efforts are also driven by the need to provide culturally competent training, which is an approach encouraged by the goals of the American Geriatrics Society statement on appropriate professional education [26]. Particularly crucial is the demand for cost-effective and evidence-based (or evidence-informed) educational initiatives to improve care of people living with dementia [25]. Therefore, evaluating cost-effective educational interventions that equip care partners with proactive, relational, and culturally responsive strategies is essential for ensuring the widespread dissemination of effective dementia care practices.

To address the need for cost-effective and culturally competent dementia training that encompasses both person- and relationship-centered approaches, one of the authors of this manuscript (Jennifer Carson), in partnership with the Nevada Department of Veterans Services and the Perry Foundation, developed and delivered Bravo Zulu: Achieving Excellence in Relationship-Centered Dementia Care (hereafter “Bravo Zulu”). Designed to integrate PCC, RCC, military cultural competence, and cultural humility within dementia care settings serving veterans, Bravo Zulu aims to enhance personhood beliefs among family and professional care partners and equip them with the skills and confidence to proactively support the needs of PLWD. This evaluation of Bravo Zulu examines its effectiveness in strengthening participants’ personhood beliefs and self-efficacy, demonstrating that dementia training rooted in PCC, RCC, cultural competency, and cultural humility can foster meaningful improvements in dementia care practices.

## 2. Materials and Methods

### 2.1. Training Development

Bravo Zulu is a 12 h dementia training program developed to enhance person- and relationship-centered care for veterans and non-veterans alike. The program’s name is derived from the naval and maritime signal “Bravo Zulu,” meaning “outstanding performance” or “job well done.” The aim is to help care partners reach “Bravo Zulu” in dementia care and support. Initially designed as part of the Veterans-in-Care initiative led by the Nevada Department of Veterans Services (NDVS), the training incorporates military cultural competence while also emphasizing broader cultural competency and humility in dementia care. While originally developed for care partners supporting veterans, the training is also applicable to care partners of non-veterans, as it promotes individualized, culturally aware, and relationship-centered care approaches.

The training was designed to fulfill three primary objectives: (1) enhance military cultural competence and humility among care partners; (2) provide comprehensive education on dementia care and support using a relationship-centered lens; and (3) apply the VIPS framework to support person- and relationship-centered dementia care. Developed by Brooker (2007), the VIPS framework builds upon Kitwood’s foundational philosophy of person-centered dementia care and delineates four essential elements: valuing the person (V), recognizing each individual’s uniqueness (I), looking at the world from the perspective of the person (P), and fostering a supportive social environment (S) [27].

Bravo Zulu consists of four 3 h modules that integrate interactive multimedia lectures, group discussions, and structured activities, offered in two live formats: in-person and online (via Zoom). Each module aligns with Brooker’s VIPS framework and expands upon each component through a relationship-centered lens. Module 1 includes Valuing Personhood, Relationships, and Culture; Module 2 focuses on Treating People as Unique Individuals; Module 3 explores Looking at the World from the Perspective of the Person; and Module 4 emphasizes Providing a Positive and Supportive Social Environment. An overview of these modules, including specific themes discussed, is provided in Figure 1.

While Bravo Zulu consistently incorporates veterans and military culture as central examples, again, its core concepts apply broadly to dementia care and support. Care partners who complete the training develop a deeper understanding of personhood, relationships, and cultural influences, equipping them to provide dementia care that is compassionate, respectful, proactive, and tailored to each individual’s strengths, preferences, and needs. The Appendix A provides the detailed structure, the concepts used, and the specific objectives of each module.

A series of training sessions were conducted in two formats: (1) in-person sessions held over two consecutive days (two modules per day) and (2) virtual sessions via Zoom, delivered weekly over four 3 h sessions. All training sessions, open to anyone interested in learning about dementia and relationship-centered approaches, were provided at no cost to the participants through grant funding. The participants were eligible to receive 12.0 Continuing Education Units (CEUs) upon completion of the training, as approved by the Board of Examiners for Long Term Care Administrators, Marriage Family Therapists, and Social Workers.

### 2.2. Training Evaluation

The results presented in this manuscript are based on findings from 11 training sessions held between September 2020 and May 2023, with participant numbers ranging from 6 to 42 per training. A total of 233 participants successfully completed all four modules of the Bravo Zulu training. To gauge the training’s impact, self-administered questionnaires were distributed during registration and immediately after the training. Approval of secondary data analysis (i.e., evaluation of training impact) was obtained from the University of Nevada, Reno Institutional Review Board (IRB No. 2258703-1).

We opted for a traditional pre- and post-assessment approach for evaluation to mitigate potential overestimation of program-related changes. Given that Bravo Zulu is a 12 h training program, there is a higher likelihood of forming positive relationships with the series facilitators, which could potentially skew the assessment of program-related changes [28]. Out of the 233 participants who engaged in training sessions held from September 2020 to May 2023, 203 completed the program evaluation, resulting in a 90% response rate. Following data cleaning—excluding participants who took the training multiple times or had missing values for key variables—a total of 182 participants with complete data were included in this study’s analysis.

Two primary outcomes were assessed to determine the impact of the training: change in personhood beliefs and self-efficacy in dementia care. Personhood beliefs were assessed using the Personhood in Dementia Questionnaire (PDQ), a 20-item tool gauging beliefs about personhood in dementia [29]. Stronger positive beliefs were associated with an increased likelihood that healthcare providers or care partners would choose a person-centered approach, taking into account an individual’s unique perspective [29,30]. Responses were recorded on a 5-point Likert scale ranging from 1 (strongly disagree) to 5 (strongly agree). To align with the training audience, we modified the term “Residents with dementia” to “Individuals living with dementia.” In the current study, the PDQ instrument shows good internal consistency, with a Cronbach alpha coefficient 0.90.

Self-efficacy, a key factor in how individuals respond to care-related stressors, was measured using five statements adapted from the Caregiving Self-Efficacy Questionnaire [31]. Bravo Zulu encourages care partners to reframe dementia using a social and relational perspective, viewing so-called “behaviors” as expressions of distress and/or unmet needs. As such, we specifically chose “disruptive behaviors” as a subscale domain, given that a person-centered response to such expressions of distress is pivotal in valuing the individual and achieving a positive outcome [31,32]. Responses were recorded on a 0–100 scale, with 0 representing “not at all confident,” 100 representing “totally confident,” and 50 indicating moderate confidence. The five statements on responding to “disruptive behaviors” have an internal consistency of Cronbach alpha of 0.94.

### 2.3. Statistical Analysis

All analyses were conducted using SPSS version 29. Sample characteristics were analyzed using descriptive statistics. Changes in participants’ personhood beliefs and self-efficacy were assessed through paired samples t-tests. Although the pre- and post-training scores exhibited moderate skewness and kurtosis, we conducted a paired samples *t*-test due to the relatively large sample size (*N* = 182), which provides robustness to violations of normality under the Central Limit Theorem [33]. Additionally, prior research supports the use of parametric tests for Likert-type data even when distributions are non-normal or unequal, without substantially increasing the risk of incorrect conclusions [34,35]. An aggregated personhood belief score was calculated by reverse-coding negatively worded items and summing the ratings across all 20 statements of the PDQ. Participants with no missing values for all 20 items were included in the pre- and post-assessment of personhood beliefs. A mean self-efficacy score was calculated by averaging participants’ ratings across five statements related to responding to so-called “disruptive behaviors.” Only participants with complete data (i.e., no missing values on all five statements at pre- and post-level) were included in the paired sample *t*-test analysis.

## 3. Results

### 3.1. Participants Characteristics

Among the 182 participants who completed both pre- and post-evaluations, the mean age was 48 (SD 13.48), with 87% being female. Additionally, 27% of participants were providing care to individuals in a rural setting. Overall, 74% of participants had formal care partner training. The training participants’ sample characteristics are shown in Table 1.

### 3.2. Evaluation of Training Impact: Personhood Beliefs and Self-Efficacy

The paired-samples *t*-tests demonstrated significant improvements in personhood beliefs and self-efficacy following Bravo Zulu training (see Table 2). The participants exhibited a statistically significant increase in personhood beliefs, with a mean pre-test score of 87.12 (SD = 10.64), rising to 92.97 (SD = 7.18) post-test. Self-efficacy also showed a significant increase, with a mean pre-test score of 86.20 (SD = 18.78), increasing to 92.74 (SD = 10.49) post-test. Despite a ceiling effect observed at registration, where 75% of participants scored above 81 in personhood beliefs and 75% of participants scored above 79 in self-efficacy, the training effectively addressed this effect, resulting in statistically significant improvements. The mean change in personhood beliefs was 5.85 (SD 9.19), 95% CI [4.51, 7.20)], indicating a medium effect, while the mean change in self-efficacy was 6.53 (SD = 16.13), 95% CI [4.71, 8.90], indicating a small effect.

As the pre- and post-training scores exhibited moderate skewness and kurtosis, we performed a non-parametric Wilcoxon Signed-Rank Test to further validate our findings, which produced results consistent with those of the paired *t*-test. Specifically, for personhood beliefs, *Z* = 8.270, *n* = 182, *p* < 0.001, with a medium effect size (*r* = 0.43) (pre-median score = 89.5, post-median score = 95); and for self-efficacy, *Z* = 4.918, *n* = 181, *p* < 0.001, with a small effect size (*r* = 0.26) (pre-median score = 94, post-median score = 96.2). These results support the robustness and reliability of our conclusions.

Further analysis examined whether the mode of delivery (in-person vs. online) had an impact on changes in personhood beliefs and self-efficacy. The results indicated that there were no significant differences in gains based on training format for either personhood beliefs *(t*(180) = −1.729, *p* = 0.09) or self-efficacy (*t*(179) = −0.525, *p* = 0.60). These findings underscore the effectiveness of the Bravo Zulu training in enhancing personhood beliefs and self-efficacy in responding to expressions of distress and unmet needs, regardless of the mode of delivery.

Given the ceiling effects observed in both personhood beliefs and self-efficacy within our study sample, we hypothesized that individuals with prior healthcare experience or formal care partner training before participating in Bravo Zulu would show higher baseline scores and smaller changes in personhood beliefs and self-efficacy following the training. To address these questions, we conducted a two-way ANOVA to examine interaction effects rather than conducting separate tests to reduce the risk of type I error [36]. Descriptive statistics for changes in personhood beliefs and self-efficacy resulting from the training are presented in the Table 3.

The two-way ANOVA analysis revealed no interaction effects between having a healthcare professional background and formal care partner training on changes in personhood belief (*F*(1, 176) = 1.300, *p* = 0.256) or self-efficacy *(F*(1, 175) = 0.021, *p* = 0.885). Furthermore, the main effect analysis for healthcare professional background or prior care partner training did not show a statistically significant effect on personhood belief or self-efficacy. However, in the case of personhood beliefs, it was observed that healthcare professionals without prior care partner training experienced greater gains (*M* = 10.23, *SD* = 7.09) compared to other groups, although the main effect was not significant (*p* = 0.056). This suggests that Bravo Zulu helped to narrow gaps in personhood beliefs between those with and without prior care partner training, although complete alignment was not achieved.

Further analyses explored whether the training’s impact varied based on participants’ care partner experience. Change scores were compared between participants with care partner experience with PLWD (*n* = 75) and those without such experience (*n* = 105). The results showed no significant difference in changes in personhood beliefs between the two groups (*p* = 0.964). However, changes in self-efficacy differed significantly based on care partner experience (*p* = 0.010). Follow-up paired *t*-tests revealed that only participants without care partner experience showed a significant increase in self-efficacy following the training, whereas the increase among care partners was not statistically significant. A summary of these results is presented in Table 4.

A systematic review of dementia education for informal care partners noted that care partner roles often differ by gender; however, there is limited evidence on whether the impact of dementia education varies by gender [37]. To address this gap, we examined whether changes in personhood beliefs and self-efficacy differed by gender among participants who identified as care partners to PLWD. While the results showed improvement in self-efficacy among participants overall, there was a significant gender difference detected, *t*(73) = −3.34, *p* = 0.001, Cohen’s d = −0.51, equal variances not assumed. Specifically, male participants with prior care partner experience (*n* = 10, *M* = −3.22, *SD* = 3.16) reported a decrease in self-efficacy, while female participants (*n* = 65, *M* = 3.91, *SD* = 14.89) reported an increase. Although the difference in self-efficacy change by gender was statistically significant, the small number of male participants with care partner experience (*n* = 10) warrants caution in interpretation. In contrast, no significant gender difference was observed in changes in personhood beliefs, *t*(73) = −0.18, *p* = 0.854.

We observed a significant difference in personhood beliefs by race/ethnicity at both pre-test level, *F*(3, 138) = 4.39, *p* = 0.006, η^2^ = 0.09 (a medium effect) and the post-test level, *F*(3, 138) = 3.88, *p* = 0.011, η^2^ = 0.08 (a medium effect). However, the changes as a result of training were not statistically significant by race, *F*(3, 138) = 2.54, *p* = 0.059 nor were the changes in self-efficacy, *F*(3, 137) = 1.06, *p* = 0.368. While racial differences in personhood beliefs persisted after the training, the training appeared to reduce these differences, as illustrated by a flatter distribution at the post-test compared to pre-test (Figure 2). Nonetheless, the training did not fully eliminate race- or culture-based biases. A more detailed analysis of racial differences in personhood beliefs is presented in a separate publication.

## 4. Discussion

The findings from this evaluation study demonstrate that the Bravo Zulu training program successfully increased both personhood beliefs and self-efficacy among program participants. Previous research has shown that strong personhood beliefs among care partners are associated with improved quality of life for PLWD in long-term care settings [38]. Additionally, higher levels of personhood beliefs have been linked to increased job satisfaction and reduced stress among healthcare professionals and other care partners working in dementia care settings [39]. Similarly, a positive attitude toward individuals living with dementia was found to be a strong predictor of a sense of competence in dementia care [40], and it increased shared decision-making in dementia care [30]. Given these findings, the significant increase in personhood beliefs observed in this study suggests that the Bravo Zulu training may contribute to improved quality of care, well-being, and job satisfaction among care partners, thus leading to higher quality of life among people living with dementia

A systematic review examining healthcare students’ attitudes toward “patient-centered” [or person-centered] care found that many students exhibit low attitudes with regard to such principles [41]. Our study provides evidence that a 12 h training intervention, such as Bravo Zulu, can significantly increase personhood beliefs among healthcare professionals without prior training. Given the well-documented association between personhood beliefs on job satisfaction [39], sense of competence [40] and shared decision making in dementia care [30], we recommend adopting training structures similar to those outlined in the Bravo Zulu training or providing training opportunities specifically designed to enhance beliefs about person- and relationship-centered approaches. Expanding access to such training programs may also address workforce challenges in dementia care, including staff retention, a pressing issue in long-term care [42].

Recent research by Scerbe et al. (2023) found no evidence supporting the effectiveness of technology-based dementia education in improving informal care partners’ self-efficacy [37]. In contrast, Bravo Zulu, which was delivered primarily via Zoom (92% of participants), successfully increased self-efficacy among the total participant sample. Further, while it was not statistically significant, there was an increase in self-efficacy among the sub-group of care partners. This suggests that Bravo Zulu’s live, interactive, and relationship-centered approach may be an effective pedagogical method, as it facilitates real-time engagement and encourages participants to share their experiences. Although care partners of persons living with dementia did not experience the same level of self-efficacy gains as those who were not providing direct support, our findings suggest that Bravo Zulu remains an effective intervention for enhancing self-efficacy among care partners overall. However, there may be a need for more targeted training structures to better support care partners actively engaged in supporting persons living with dementia. The same review also highlighted the need to better understand how gender differences in care partner roles may influence the impact of dementia education [37]. Our findings offer preliminary evidence that gender may play a role: male participants with care partner experience (n = 10) did not show a significant increase in self-efficacy following the training, while female participants did. However, this result should be interpreted with caution due to the small number of male participants with care partner experience. Future research should investigate gender differences in training outcomes using more balanced samples.

These results may indicate opportunities for more inclusive training approaches that reflect the diverse population of care partners for PLWD. As the U.S. healthcare workforce becomes increasingly diverse, culturally competent training programs such as Bravo Zulu have the potential to reduce racial differences in personhood beliefs and improve care outcomes. Notably, Bravo Zulu’s live, interactive, and relationship-centered design appears especially effective for individuals without prior care partnering experience and for female participants, regardless of care partnering status.

There is no single approach sufficient to fully recognize the rights of PLWD, understand their evolving identities, honor cultural differences, and respond to their complex needs [16]. Healthcare professionals require guidance that integrates biological, psychological, and social factors to deliver truly person- and relationship-centered care [16]. The Bravo Zulu training curriculum addresses this need by offering an integrative framework that emphasizes personal identity, cultural context, and reciprocal well-being among people living with dementia and care partners. It supports understanding dementia as a co-constructed experience shaped by relationships and social environments. Drawing on Kitwood’s [43] emphasis on preserving the identity of PLWD and Nolan’s [11] framework of the six ‘senses’ in care (security, belonging, continuity, purpose, achievement, and significance), Bravo Zulu is uniquely designed to foster both high levels of personhood beliefs and self-efficacy among care partners. This unique approach bridges clinical and relational knowledge, preparing care partners to support PLWD with empathy, dignity, and cultural competence as dementia progresses.

As the field of dementia care faces increasing demand for transformative, cost-effective, scalable training interventions [25], Bravo Zulu demonstrates its potential as an effective, accessible, and flexible training model. The program’s ability to produce significant increases in personhood beliefs and self-efficacy, regardless of the mode of delivery (in-person vs. online), underscores its adaptability. Furthermore, Bravo Zulu’s online format option allows for broader dissemination, particularly in rural areas where access to dementia training is often limited. By overcoming geographical barriers, the program serves as a valuable tool for strengthening relationship-centered dementia care capacity across diverse care settings.

Despite the positive impact of the training, this study has some important limitations. First, post-training assessments were conducted immediately after the training, meaning that we did not measure the long-term sustainability of changes in personhood beliefs and self-efficacy. Future research should include longitudinal follow-up assessments to determine whether these effects persist over time. Second, while this study focused on participant-level outcomes, we did not examine how training impacted the experiences, well-being, or quality of life of PLWD or care partners. Third, we did not include a comparison group—such as a control or an alternative intervention group, which limits our ability to determine whether the observed changes in participants’ personhood beliefs and their self-efficacy in responding to expressions of distress or unmet needs can be attributed specifically to the Bravo Zulu training. Future studies should assess whether increases in personhood beliefs and self-efficacy translate into tangible improvements in dementia care practices and outcomes for both groups.

This study contributes to the growing body of research supporting the effectiveness of relationship-centered dementia education and highlights the need for continued innovation in training models. Future research should explore how Bravo Zulu and similar training programs can be optimized for different care settings, integrated into dementia workforce development, and tailored to meet the needs of diverse populations, including experienced care partners, rural providers, and underrepresented communities.

## 5. Conclusions

Overall, the Bravo Zulu training significantly improved participants’ personhood beliefs, as well as their self-efficacy in responding to expressions of distress and/or unmet needs, demonstrating its value as an effective and scalable dementia care training model. While the distal outcome of improvements in quality of care were not assessed in this evaluation, our theory of change for this intervention assumes that enhanced personhood beliefs and self-efficacy among care partners will result in better quality of care for people living with dementia. Healthcare professionals without prior care partner training derived greater benefits from Bravo Zulu compared to non-healthcare professionals or healthcare professionals with prior training. However, while the training was effective, it did not fully eliminate the influence of prior care partner training experiences, suggesting that ongoing education may be essential for sustaining high levels of personhood beliefs among professional healthcare providers and other care partners. This study underscores the importance of continuous dementia care training, particularly in addressing ceiling effects and recognizing variations in training benefits based on participants’ professional backgrounds (healthcare or non-healthcare), gender, race, and care partner experience. Tailoring dementia education to diverse audiences is crucial for ensuring the adoption of effective, relationship-centered care practices that enhance the well-being of both care partners and PLWD.

To meet the growing demand for high-quality dementia care, organizations and policymakers should invest in training that goes beyond traditional, didactic approaches and instead emphasizes relationship-centered, interactive learning. Programs such as Bravo Zulu demonstrate that structured, culturally attuned, and engaging dementia education can significantly enhance care partners’ personhood beliefs and self-efficacy, ultimately improving the experience of care for PLWD. Organizations and policymakers can support the widespread adoption of such training by integrating relationship-centered approaches into workforce development initiatives, incentivizing continuing education and ensuring accessibility—particularly in underserved and rural communities. By prioritizing culturally competent and relationship-centered practices within dementia training, the field of dementia care can move toward a more compassionate, reciprocal, and fulfilling future that supports the well-being of people living with dementia.

## Figures and Tables

**Figure 1 ijerph-22-00970-f001:**
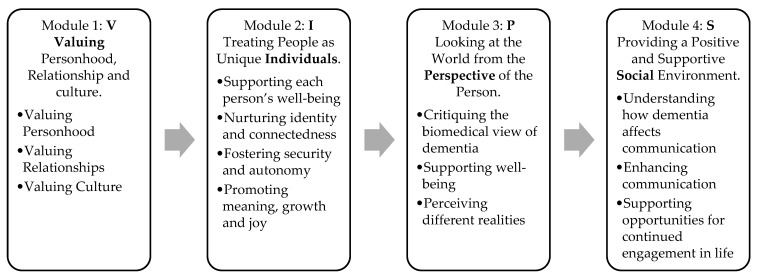
An overview of the training program, including specific themes discussed in each module.

**Figure 2 ijerph-22-00970-f002:**
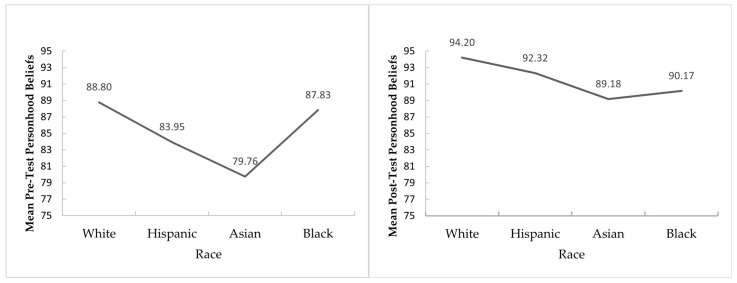
Pre-and post-test mean difference in personhood beliefs based on race/ethnicity.

**Table 1 ijerph-22-00970-t001:** Demographic profile of Bravo Zulu training participants (*N* = 182).

Characteristics	Number	Percentage %
Gender		
Male	23	13
Female	158	87
Missing gender	1	<1
Race/ethnicity		
White	108	48
Hispanic	22	12
Black	23	13
Asian	24	13
Missing race/ethnicity	5	3
Primary role		
Health professional/provider	53	29
University/college faculty	3	2
Other (non-healthcare related)	29	16
Healthcare administration	7	4
Public health (e.g., health program director, evaluation, and/or worker)	5	3
Social work	36	20
Missing primary role	49	27
First generation college students	87	48
Previously attended any formal care partner * training	135	74
Worked as healthcare professional **	105	58
Care partner to someone living with dementia	75	41
Caring for or serving someone who has served in the Armed Forces	41	23

Notes: * While the demographic survey used to collect required data for the HRSA-funded Geriatrics Workforce Enhancement Program included the term “caregiver”, throughout this paper we use our preferred terminology of “care partner”, as included in the table above. Operationally, in the data collection tool, “caregiver” was defined as anyone who provides care and/or assistance to another individual, either professionally in an assisted living facility, nursing home, or home health setting OR within the community, including care given to family members, friends, and/or neighbors. ** “Healthcare professional” was defined in the survey as someone who has worked in the healthcare industry as either a doctor, nurse, nurse’s aide, or other direct care worker.

**Table 2 ijerph-22-00970-t002:** Results of paired *t*-test examining the training impact.

Variable	Pre		Post		*t*-Statistics	Cohen’s d
	*M*	*SD*	*M*	*SD*		
Personhood Beliefs	87.12	10.64	92.97	7.18	*t*(181) = 8.586, *p* < 0.001	0.63
Self-Efficacy	86.20	18.78	92.74	10.49	*t*(180) = 5.449, *p* < 0.001	0.40

**Table 3 ijerph-22-00970-t003:** Descriptive statistics for change in personhood beliefs and self-efficacy.

Variable	Profession	Training	*M*	*SD*	N
Personhood Beliefs	Non-healthcare professional	No training	6.08	11.14	24
		Prior care partner training	4.86	9.10	51
	Healthcare professional	No training	10.23	7.06	22
		Prior care partner training	5.45	8.85	83
Self-Efficacy	Non-healthcare professional	No training	7.18	20.90	24
		Prior care partner training	9.31	17.59	51
	Healthcare professional	No training	4.13	11.99	22
		Prior care partner training	5.44	14.71	82

**Table 4 ijerph-22-00970-t004:** Comparison of training impact on personhood beliefs and self-efficacy based on care partnering experience.

Variable	Care Partners		Non-Care Partners		*t*-Statistics	Cohen’s d
	*M*	*SD*	*M*	*SD*		
Change in Personhood Beliefs	5.99	8.91	5.92	9.34	*t*(178) = 0.045, *p* = 0.964 ^a^	n/a
Change in Self-Efficacy	2.96	14.11	9.25	17.14	*t*(177) = −2.603, *p* = 0.010 ^a^	0.39
	**Pre**		**Post**		***t*-statistics**	**Cohen’s d**
**Non-Care Partners**	*M*	*SD*	*M*	*SD*		
Personhood Beliefs	86.57	10.77	92.50	7.01	*t*(104) = 6.500, *p* < 0.001 ^b^	0.63
Self-Efficacy	84.39	17.69	93.08	8.26	*t*(90) = 5.125, *p* < 0.001 ^b^	0.54
**Care Partners**						
Personhood Beliefs	87.79	10.61	93.77	7.01	*t*(74) = 5.820, *p* < 0.001 ^b^	0.67
Self-Efficacy	88.78	18.18	91.88	13.38	*t*(180) = 5.449, *p* = 0.069 ^b^	n/a

^a^ independent *t*-test statistics; ^b^ paired *t*-test statistics.

## Data Availability

The raw data supporting the conclusions of this article will be made available by the authors on request.

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
