# Peer review of "Increasing Care Partners’ Capacity for Supporting Individuals Living with Dementia Through Bravo Zulu: Achieving Excellence in Relationship-Centered Dementia Care"

_ijerph, 2025, doi:10.3390/ijerph22070970_

Round 1

Reviewer 1 Report

Comments and Suggestions for Authors

Manuscript Title: Increasing Care Partners’ Capacity for Supporting Individuals Living with Dementia: Bravo Zulu, Achieving Excellence in Relationship-Centered Dementia Care

Recommendation: Accept with Minor Revisions

Reviewer Summary:

This manuscript offers a thoughtful, well-structured, and empirically grounded evaluation of the “Bravo Zulu” training program, an innovative educational intervention designed to increase cultural competence, personhood beliefs, and self-efficacy among care partners of individuals living with dementia. The authors integrate robust conceptual frameworks—particularly person-centered and relationship-centered care (PCC and RCC)—with a practical, scalable curriculum grounded in lived experience and rigorous training delivery.

The study design is methodologically sound, utilizing paired t-tests and ANOVA models to analyze pre- and post-intervention data from a sizable sample (N = 182). The use of validated instruments (PDQ and self-efficacy measures) strengthens the findings, which show significant improvement in both key outcomes across participant subgroups. Importantly, the manuscript addresses persistent challenges in the dementia care workforce, including disparities by race, gender, and prior caregiving experience, and makes a strong case for training accessibility through online delivery formats.

Strengths:

  • Clearly articulated rationale for transitioning from PCC to RCC. This was done nicely.

  • Empirically supported evaluation of a culturally competent training model. Again, this was done very well.

  • High relevance to public health, long-term care, and dementia education policy.

  • Accessible, flexible training model with clear potential for scale-up. This is very important and investigators often omit this in behavior science applied research.

  • Balanced discussion of limitations, including ceiling effects and underrepresentation of male participants. I really appreciate that you were very honest about your limitations and didn't try to bury them, as so many other studies attempt to do.

Minor Revisions Suggested (small, really)

  • Edit for clarity and remove minor redundancies (e.g., repetitive phrasings such as “additionally, additionally”), which was at line 358, I believe.

  • Provide a brief explanation of the VIPS framework for readers unfamiliar with dementia-specific care models. This is not required but might be helpful.

  • Maybe... standardize the formatting of tables to align with journal style guidelines, but honestly, that's it. I appreciate this article.

 __________________________

Overall, I think these small changes can be handled without a major. resubmission.

Author Response

Thank you very much for the helpful reviews provided by both reviewers. We have addressed each comment and are submitting both a tracked-changes version and a clean version of the revised manuscript. For clarity, we have described the changes made in response to each comment directly following the respective reviewer remark.

Comment 1: Edit for clarity and remove minor redundancies (e.g., repetitive phrasings such as “additionally, additionally”), which was at line 358, I believe.

Response 1: We carefully reviewed and refined the text to: 1) correct typographical errors; 2) address inconsistencies in dementia-related terms/content (i.e., caregiver versus care partner and workshop versus training ); and 3) increase contextual clarity to achieve continuity between results, discussion and conclusions.”

Comment 2 Provide a brief explanation of the VIPS framework for readers unfamiliar with dementia-specific care models. This is not required but might be helpful.

Response 2: Thank you for your feedback. We have provided additional information on the VIPS framework, as outlined below.

Developed by Brooker (2007), the VIPS framework builds upon Kitwood’s foundational philosophy of person-centered dementia care and delineates four essential elements: Valuing the person (V), recognizing each Individual’s uniqueness (I), looking at the world from the Perspective of the person (P), and fostering a supportive Social environment (S) (Brooker, 2007).

Comment 3: Maybe... standardize the formatting of tables to align with journal style guidelines, but honestly, that's it. I appreciate this article. __________________________

Response 3: Thank you for pointing this out. We have updated the table formatting accordingly and will cross-check all formatting again prior to final approval for publication.

Reviewer 2 Report

Comments and Suggestions for Authors

Hello, thank you for sharing this article, "Increasing Caregivers' Capacity to Support People with Dementia: Bravo Zulu, Achieving Excellence in Relationship-Centered Dementia Care." I really enjoyed the introduction, as it was clear and demonstrated the importance of the study. Only validated instruments were used to ensure comparability with other studies. The objective was to evaluate the Bravo Zulu intervention and examine its effectiveness in strengthening participants' personal beliefs and self-efficacy. Some important clarifications are in order:
1. The intervention program is adequately described, making it a relevant option in the study of dementia caregiving. 
2. In the statistical analysis, they do not explain how the distribution of variables was assessed to use parametric tests in the bivariate analysis.
3. The limitations of this paper do not mention the lack of a comparison group without treatment or other intervention to determine whether Bravo Zulu training improved participants' beliefs about relationship-centered care, as well as their self-efficacy in responding to expressions of distress and/or unmet needs.

Furthermore, the paper highlights the needs of people with dementia and their caregivers for training in quality person-centered care and support, with an innovative approach to care.
The conclusions are consistent with the results obtained, address the stated objectives, and answer the initial question.
Finally, the paper's references are adequate; there are few current ones (from 2019 to 2025), but most are focused on the research topic.

Author Response

Thank you very much for the helpful reviews provided by both reviewers. We have addressed each comment and are submitting both a tracked-changes version and a clean version of the revised manuscript. For clarity, we have described the changes made in response to each comment directly following the respective reviewer remark.

Comments 1. The intervention program is adequately described, making it a relevant option in the study of dementia caregiving. 

Response 1: Thank you.

Comments 2. In the statistical analysis, they do not explain how the distribution of variables was assessed to use parametric tests in the bivariate analysis

Response 2: Thank you for the valuable feedback. We have added our rationale for conducting parametric tests and, additionally, performed a Wilcoxon Signed Rank Test to confirm our results. The following information has been included in the manuscript to address the reviewer’s concern.

Although the pre- and post-training scores exhibited moderate skewness and kurtosis, we conducted a paired samples t-test due to the relatively large sample size (n = 182), which provides robustness to violations of normality under the Central Limit Theorem (Field, 2013). Additionally, prior research supports the use of parametric tests for Likert-type data even when distributions are non-normal or unequal, without substantially increasing the risk of incorrect conclusions (Carifio & Perla, 2008; Norman, 2010).

As the pre- and post-training scores exhibited moderate skewness and kurtosis, we performed a non-parametric Wilcoxon Signed Rank Test to further validate our findings. which produced results consistent with those of the paired t-test. Specifically, for personhood beliefs, Z = 8.270, n = 182, p < .001, with a medium effect size (r = .43) (Pre Md=89.5, post Md= 95) ; and for self-efficacy, Z = 4.918, n = 181, p < .001, with a small effect size (r = .26) (Pre Md =94, Post Md = 96.2)  . These results support the robustness and reliability of our conclusions.

Additionally, we would like to share detailed information on the distribution of the pre- and post-training scores in response to the reviewer’s request. This information is provided here for the review process only and is not included in the manuscript. Since both the parametric and non-parametric tests produced consistent results, we chose not to add these details to the manuscript to avoid potential confusion for readers.

The pre-training personhood belief scores exhibited a negative skew (−1.29) and a kurtosis of 2.08, with values ranging from 48 to 100, a mean of 87.11 (SD = 10.64), and a 5% trimmed mean of 88.08. The post-training scores also showed a negative skew (−0.99) and a kurtosis of −0.12, with a range of 73 to 100, a mean of 92.97 (SD = 7.18), and a 5% trimmed mean of 93.51. The pre-training self-efficacy scores were more strongly negatively skewed (−2.02) with a kurtosis of 5.03, ranging from 0 to 100, a mean of 86.20 (SD = 18.78), and a 5% trimmed mean of 88.43. The Post -training self-efficacy scores were more strongly negatively skewed (−2.60) with a kurtosis of 9.32, ranging from 31.40 to 100, a mean of 92.74 (SD = 10.49), and a 5% trimmed mean of 94.18.

Comment 3. The limitations of this paper do not mention the lack of a comparison group without treatment or other intervention to determine whether Bravo Zulu training improved participants' beliefs about relationship-centered care, as well as their self-efficacy in responding to expressions of distress and/or unmet needs.

Response 3: Thank you for the valuable feedback. We included additional information discussing this limitation.

Third, we did not include a comparison group—such as a control or an alternative intervention group—which limits our ability to determine whether the observed changes in participants' personhood beliefs and their self-efficacy in responding to expressions of distress or unmet needs can be attributed specifically to the Bravo Zulu training.

References

Carifio, J., & Perla, R. (2008). Resolving the 50-year debate around using and misusing Likert scales. Medical Education, 42(12), 1150–1152. https://doi.org/10.1111/j.1365-2923.2008.03172.x

Field, A. (2013). Discovering Statistics using IBM SPSS Statistics. In: Sage Publications Ltd.

Norman, G. (2010). Likert scales, levels of measurement and the "laws" of statistics. Advance in Health Science Education, 15(5), 625–632. https://doi.org/10.1007/s10459-010-9222-y